# Assessing the Influence of Stimulatory Feeding of Bee Colonies on Mineral Composition and Antioxidant Activity of Bee Venom

**DOI:** 10.3390/insects16040423

**Published:** 2025-04-17

**Authors:** Adrian Dan Rășinar, Isidora Radulov, Adina Berbecea, Doris Floares (Oarga), Nicoleta Vicar, Eliza Simiz, Monica Dragomirescu, Silvia Pătruică

**Affiliations:** 1Faculty of Bioengineering of Animal Resources, University of Life Sciences “King Mihai I” from Timișoara, Calea Aradului No. 119, 300645 Timisoara, Romania; adrian.rasinar@usvt.ro (A.D.R.); elizasimiz@usvt.ro (E.S.); monicadragomirescu@usvt.ro (M.D.); silviapatruica@usvt.ro (S.P.); 2Faculty of Agriculture, University of Life Sciences “King Mihai I” from Timișoara, Calea Aradului No. 119, 300645 Timişoara, Romania; doris.oarga@usvt.ro (D.F.); nicoleta.vicar@usvt.ro (N.V.)

**Keywords:** bee venom, chemical composition, antioxidant activity, nutritional factors

## Abstract

Bee venom is a natural product traditionally used in therapy and known for its health benefits. However, little research has focused on its mineral content, which varies based on the environment. Our study analyzed the mineral composition of bee venom and investigated how different supplements, such as probiotics, essential oils, and various nectar and pollen sources, influence its properties. We measured moisture, pH, mineral levels, and antioxidant activity. The results showed that potassium was the most abundant mineral, followed by calcium, magnesium, and phosphorus. These findings help improve the understanding of bee venom composition.

## 1. Introduction

For thousands of years, natural bee products, including bee venom [1,2,3], have been used therapeutically in apitherapy, a significant branch of alternative medicine [4,5,6]. Worker bees secrete venom from their venom glands, producing between 0.05 and 0.3 mL per bee, depending on the species and the season [7]. At the moment of the sting, bee venom is a semi-transparent liquid with a pungent odor and bitter taste [8,9], and it has a pH range of 4.5 to 5.5 [10,11].

Bee venom has been extensively studied for its potential therapeutic applications in treating conditions such as arthritis, multiple sclerosis, and some types of cancer [12]. Recently, research has focused on the antioxidant activity of bee venom, which can reduce oxidative stress and promote health. In mammalian cells, vitellogenin provides protection against oxidative stress by defending against reactive oxygen species [13]. Research [14] showed that injecting rabbits with bee venom twice a week improved reproductive performance by enhancing the antioxidant capacity of sperm. Studies on rats with rheumatoid arthritis [15] and gastric ulcers [16] have also confirmed bee venom’s antioxidant properties.

Bee venom, also known as apitoxin, is a complex natural substance soluble in water but insoluble in alcohol [8], comprising over 214 different components [17]. Water is the primary constituent, accounting for approximately 88% of the venom’s composition [18]. It contains a variety of bioactive compounds, including peptides like melittin, apamin, mast cell degranulating peptide (MCDP), and adolapin; proteins with enzymatic functions such as phospholipase A2, hyaluronidase, and phosphatase; as well as amines, volatile substances, carbohydrates, alcohols, sugars, and hydrocarbons [3,8,19,20,21,22,23].

Bee venom also contains 3–4% minerals that are crucial for its biological activity. Key minerals include potassium (K), calcium (Ca), and magnesium (Mg), which contribute to the venom’s overall efficacy and therapeutic potential. These minerals, along with other bioactive compounds, contribute to bee venom’s anti-inflammatory, anti-microbial, and pain-relieving properties [19,24,25,26,27,28]. However, bee venom can accumulate heavy metals, such as lead (Pb), cadmium (Cd), nickel (Ni), copper (Cu), and zinc (Zn) [29], due to environmental contamination from polluted plants, water, and air.

The health of bee colonies directly impacts the quantity and quality of beekeeping products, and consequently the quantity and quality of bee venom. Healthy bee colonies can be maintained with probiotics and prebiotics [11,30], as well as essential oils [31]. The amount of bee venom produced can be influenced by the diet of bee colonies [11,32], while its quality is affected by environmental factors, harvesting time and method [33,34], bee age [35], supplementary feeding [36], the treatments that were applied to the crops [37], and season [38].

Recent studies indicate that essential oils (EOs) hold promise as alternatives to antibiotics due to their antioxidant, antibacterial, and antiviral properties [39,40,41,42]. Research has shown that combining honey with EO can enhance antimicrobial efficacy. Adding essential oils like oregano and mint to bee feed has been linked to a reduction in intestinal pathogens and an increase in honey production [43]. Among the most used EOs for bee colonies are thyme, lemongrass, and spearmint oils, which are typically administered in sugar syrup or patty form. In vitro assays have demonstrated that thyme oil and its constituents can inhibit the growth of various fungal and bacterial pathogens affecting honey bees [44].

Studies have shown that feeding honey bee colonies with a probiotic mixture containing *L. acidophilus*, *L. casei*, and *Bifidobacterium lactis* significantly reduces the number of harmful gut bacteria, resulting in a higher percentage of beneficial lactic acid bacteria compared to controls [45]. In addition to supporting digestive function, pro-biotics have been shown to maintain microbial balance, strengthen the immune system, and prevent the colonization of the intestines by pathogenic spores, which can be more prevalent under conditions of stress or illness [46,47].

The mineral composition of bee venom varies depending on factors such as bee species, season, and geographical location. Research on the mineral content of bee venom is limited and challenging to compare across studies due to the varying regions where they are conducted. Therefore, our study aims to assess the mineral content of bee venom and how supplemental feeding of bee colonies with probiotic products, essential oils, as well as rapeseed and acacia nectar and pollen affects the mineral content and antioxidant activity of the venom.

## 2. Materials and Methods

### 2.1. Samples of Bee Venom

We analyzed 24 bee venom samples (*Apis mellifica carpatica* species) from 3 harvests carried out in April, May, and June 2024. The samples came from Timiș County, Romania. For statistical validation, 3 independent hives were taken into account for each experimental variant.

The first harvest (8 samples) was carried out after feeding the bee colonies for 3 weeks with sugar syrup ratio 1:1 (1 kg sugar: 1 L water), with addition of essential oils and probiotic products, according to Table 1.

The essential oils used in the study were purchased from Bioinovativ (https://life-bio.ro/produse/uleiuri-esentiale/page/6/, accessed on 26 February 2025). They were obtained through steam distillation and are certified by the Romanian National Service for Medicinal, Aromatic Plants and Beehive Products. The oils included thyme whole essential oil, wild oregano/sorrel whole essential oil, and basil whole essential oil. We used the following probiotic products: Lacium (50 billion CFU) composed of 9 strains of lactobacilli and bifidobacteria 162 mg and inulin 140 mg (*Bifidobacterium breve* BR03, *Lactobacillus rhamnosus* LR02, *Lactobacillus plantarum* LP09, *Lactobacillus casei* LC03, *Lactobacillus acidophilus* LA02, *Streptococcus thermophilus* YA08, *Bifidobacterium bifidum* BB01, *Bifidobacterium longum*, *Lactococcus lactis* LL02); Colobiotic, composed of 8 strains of lactobacilli and bifidobacteria 200 mg (*Bacillus coagulans*, *Bacillus subtilis natto*, *Lactobacillus acidophilus*, *Lactobacillus casei*, *Lactobacillus rhamnosus*, *Lactobacillus plantarum*, *Bifidobacterium bifidum*, *Bifidobacterium breve*), and Enterolactis Plus, which contains *L. casei* 250 mg (*Lactobacillus paracasei* CNCM I-1572).

The second harvest (8 samples) was carried out after the rapeseed harvest, and the third harvest after the acacia harvest (8 samples).

Venom collection was performed using the BeeWhisper venom collector v.5.1 model 2016 (Figure 1). The venom collector operating time for one collection series was 30 min.

Analyzes were performed at the Interdisciplinary Research Platform (PCI) belonging to the University of Life Sciences “King Mihai I” from Timisoara.

### 2.2. Determination of Humidity

We determined the humidity using a thermobalance. The principle of the method consists in the continuous measurement of the sample mass during the drying process, using equipment that combines a high-precision balance and a controlled heating source. Final results are obtained directly in the form of moisture percentage. A calibrated thermobalance (Sartorius MA160, Sartorius AG, Göttingen, Germany) was used for the determination, and 0.01 g of venom was used for each analysis. The sample was placed on the thermobalance plate, and the equipment was set to a drying temperature of 100–105 °C. The determination process continued until the sample mass stabilized, indicating complete removal of moisture. Three such determinations were performed to ensure the accuracy of the results.

### 2.3. Determination of Dry Matter

Following moisture determination, the moisture and dry matter added together represent 100%. The following formula was used for the determination:Dry matter = 100 − Humidity (%)(1)

### 2.4. Determination of pH

The pH was determined by the conductometric method using a lino-Lab pH 720 pH meter (Xylem Analytics, Weilheim, Germany) [48]. For the determination, 0.003 g of venom was dissolved in 10 mL of water, homogenized for 30 min [31] with the help of a stirrer DLAB (SK-L330-PRO, Beijing, China).

### 2.5. Determination of Impurities

It was made according to the SR-784-3:2009 [49] standard as follows: for venom samples, 0.003 g/sample was weighed and dissolved in 10 mL water and homogenized using a DLAB shaker (SK-L330-PRO, China), for 30 min. Subsequently, filter papers were prepared and weighed, and after weighing them, the solutions were filtered. The samples obtained after filtration were placed in an oven at a temperature of 103 °C for 25 min to dry the filter paper, then weighed. The impurity content was expressed in percentages and was determined according to the formula:I = (m_2_/m_1_) × 100 (%)(2)
where:I—represents the quantity of impurities (%);m_1_—represents the mass of the sample taken for analysis (g);m_2_—represents the mass of residue left on the filter paper after drying (g).

### 2.6. Determination of Ash

It was carried out by the loss on ignition method according to SR 784-3:2009 [49] using an oven (190945, Nabertherm, Lilienthal, Germany). An amount of 0.1 g of sample was calcined at a temperature of 525 °C for two hours, until white-gray ash was obtained. The ash content was determined with the relationship:Cash = (m − m_1_)/(m_2_ − m_1_) (%)(3)
where:Cash = total ash contentm—the mass of the crucible with ash after the calcination process, in grams;m_1_—the mass of the empty crucible, in grams;m_2_—the mass of the crucible with the bee products, before the calcination process, in grams.

### 2.7. Determination of Total Amino Acids

To determine the amino acids, 72 lidded containers were prepared corresponding to 3 determinations for each of the 24 experimental variants. A 3 mg portion of each venom sample was weighed into each plastic container, followed by the addition of 10 mL of distilled water. The samples were homogenized using a DLAB shaker (SK-L330-PRO, Beijing, China) for 30 min, then filtered through filter paper. Subsequently, 1 mL of the resulting extract was pipetted into a glass tube for each of the 48 determinations.

Next, 0.5 mL of phosphate buffer solution (pH 8.04) and 0.5 mL of 2% ninhydrin solution were added to each test tube. These were shaken again with the DLAB shaker (SK-L330-PRO, Beijing, China) for 30 min and then placed in an oven (BINDER GmbH, Tuttlingen, Germany) at 103 ± 2 °C for 10 min. After removing the test tubes from the oven, they were each filled with distilled water up to 10 mL and manually homogenized. Finally, the samples were allowed to rest for 15 min before measuring the absorbance at 570 nm using a UV-VIS spectrometer (Analytical Jena Specord 205, Jena, Germany). A calibration curve was established using alanine standard solutions (Fluka, Madrid, Spain) with concentrations ranging from 100 to 600 µg/mL. The total free amino acid content was expressed in ppm alanine and converted to mg/g.

### 2.8. Mineral Content

#### 2.8.1. Determination of Total Metal Content

Bee venom samples were prepared for metal content determination using the dry ashing technique. The samples were placed in open crucibles and heated at 550 °C in a muffle furnace at atmospheric pressure to thermally decompose and remove organic matter. The resulting ash was dissolved in 6 N hydrochloric acid (HCl), and after filtration, the solution was diluted to a final volume of 50 mL with bidistilled water. Metal content analysis was conducted using atomic absorption spectrometry on a Varian Spectra 240 FS (Palo Alto, CA, USA) spectrophotometer. The working conditions are detailed in Table 2.

Air: acetylene ratio of 13.50:2.

Nebulizer uptake rate of 5 L/min.

For calibration standard solutions were used with the concentration ranging from 0.3 to 3 μg/L, prepared from multielement solution ICP Standard solution 1000 mg/L.

#### 2.8.2. Determination of Total Phosphorus Content

Total phosphorus content was determined from the ash dissolved in 6N HCl using the UV–Vis spectrometry method with a Cintra (Keysborough, Australia) spectrophotometer. A specific volume of the solution was treated with an acidic solution of ammonium molybdate and stannous chloride to form ammonium phosphomolybdate, which was measured colorimetrically at a wavelength of λ = 715 nm. Calibration was performed using standard solutions with concentrations ranging from 2 to 60 μg/100 mL, prepared from a multielement ICP standard solution of 1000 mg/L. All reagents used were of Merck grade.

### 2.9. Determination of the Antioxidant Capacity

The free radical neutralization activity was assessed using the DPPH (2,2-Diphenyl-1-picrylhydrazyl) method with a 0.3 mM DPPH solution in 70% ethyl alcohol. This commonly used method evaluates the antioxidant activity of samples by their ability to neutralize free radicals. A 0.010 g venom sample was extracted with 4 mL of distilled water using a Holt plate shaker (IDL, Freising, Germany) for 30 min. The samples were then filtered and stored at 4 °C until chemical analysis.

For the analysis, 1 mL of the extract was mixed with 2.5 mL of the DPPH solution. Five different concentrations of bee venom were used: 2%, 1.25%, 1%, 0.625%, and 0.313%. The mixture was shaken vigorously and incubated in the dark at room temperature for 30 min. After incubation, the absorbance was measured at 518 nm using a UV–VIS spectrophotometer (Specord 205, Analytik Jena AG, Jena, Germany). A control sample was prepared by replacing the extract with distilled water. The analysis was performed in triplicate, and the average results were reported.RSA(%) = (A control − A sample)/(A control) × 100(4)
where A control = absorption value of the control sample; A sample = absorption value of the extracts.

The antioxidant capacity of the extracts was expressed by the IC_50_ value and compared with that of ascorbic acid.

### 2.10. Statistical Analysis

The results presented in this study were determined using the IBM SPSS 22 statistical program. In the case of statistical differences (*p* < 0.05) between the bee products analyzed, they were processed using the Anova program with Tukey’s test. Pearson correlation and PCA wore perform using Origin 2025.

## 3. Results and Discussion

### 3.1. Physicochemical Characterization of Bee Venom

#### 3.1.1. Moisture, Dry Matter, Ash, Impurities, and pH

The bee venom samples we analyzed across three harvests, namely after the stimulation feeding in April (using sugar syrup, essential oils, and/or probiotic products), after the rapeseed harvest in May, and after the acacia harvest in June, exhibited significant variability in chemical composition influenced by the season and type of feeding. The moisture content of the samples was 13–18% for the first harvest, 10–22% for the second harvest, and 15–18% for the third harvest (Table 3). During the first harvest, the lowest moisture values were observed with basil essential oil (V2) and the Colobiotic probiotic product (V6), while the control group fed only with sugar syrup (V1) had the highest value. In the second harvest, the lowest moisture content was noted in sample V14, and during the third harvest, the moisture content across samples was relatively consistent (Table 3).

Our data showed higher moisture content compared to other studies, which reported ranges of 9.52–12.38% [50] and 9.16–10.56% [7]. This difference may be influenced by the atmospheric humidity at the time of venom collection. The samples were taken from hives in Timis County, Romania, located at 45 degrees 41 min N latitude, with relative air humidity of 68%, 69%, and 67% in April, May, and June, respectively. This higher humidity may account for the increased moisture content in our samples compared to areas where other bee venom humidity studies were conducted [7,50]. The dry matter content of the samples was 82–87% for the first harvest, 78–90% for the second harvest, and 82–85% for the third harvest, correlating with their moisture percentages (Table 3).

The bee venom obtained in the first harvest had a pH range of 5.84 to 6.41, in the second harvest it ranged from 5.54 to 6.11, and in the third harvest it was between 5.93 and 6.06. During the first and second harvests, the highest pH values were observed in V1 and V9, followed by V11 and the variants with essential oils, V3-V4. For the third harvest, the highest pH values were recorded in variants V24, V18, and V22. Among the highest pH values determined in harvest I (V1—6.41 ± 0.02; V3—6.10 ± 0.05; and V4—6.02 ± 0.05), there are significant differences at a statistical confidence level of α = 0.05. In harvest II, no significant differences were observed between the highest pH values recorded in V9 (6.11 ± 0.04) and V11 (6.11 ± 0.04). The same observation applies to the highest pH values in harvest III (V24—6.06 ± 0.04; V18—6.04 ± 0.04; and V22—6.03 ± 0.04), where no significant differences were found. Across all samples, the pH values were higher compared to those reported by other authors: 4.5–5.5 [51,52,53] and 5.0–5.5 [11]. Certain food supplements can influence the pH balance of honeybee venom. In our study, we used tap water with a pH between 6.6 and 6.9 to prepare sugar syrup, resulting in a syrup with a pH of 6.4. Adding probiotic supplements containing Lactobacillus and Bifidobacterium strains, with a pH around 4–6 [54], to the sugar syrup caused slight acidification. In the variants where bees were fed with these supplements, the lowest pH values (5.54 in V14) were observed, similar to those found in other studies [11,51,52,53]. The essential oils added had pH values of 5.29–6.31 for basil oil [55], 5.3–5.7 for oregano oil [56], and neutral for thyme oil. Consequently, feeding bees with these food supplements can alter the pH of bee venom, with the highest values recorded when feeding only sugar syrup and sugar syrup combined with thyme oil.

The impurities in the analyzed bee venom displayed significant variability, ranging from 6.67% to 13.33% in the first harvest, 6.67% to 20% in the second harvest, and 6.67% to 13.33% in the third harvest. Statistically significant differences were observed both between the variants within a single harvesting session and across the three harvests (Table 3). There are no established guidelines for permissible levels of impurities in bee venom, and the large differences in impurity levels between batches are likely due to contamination during the harvesting process.

#### 3.1.2. Total Amino Acid Content

Peptides and proteins constitute the majority of the dry mass of bee venom, with melittin being the primary component. Melittin, a peptide composed of 26 amino acids, accounts for approximately 40–60% of the dry weight of bee venom [10]. Other peptide components present in bee venom include apamin, mast cell degranulation peptide, adolapine, secapine, procamine, and phospholipase A2 [57].

In our analysis, the amount of amino acids in the bee venom samples ranged from 335.95 to 398.20 mg/g during the first harvest, 354.52 to 403.11 mg/g in the second harvest, and 396.02 to 407.48 mg/g in the third harvest (Table 3). Between the highest values determined for each harvest (V1—harvest I; V16—harvest II; V17—harvest III) and the other values determined within the same harvest, there are significant differences at a statistical confidence level of α = 0.05. An increase in amino acid levels was observed across the three harvests, indicating that the abundant pollen harvest in April and May positively influences the protein quality of bee venom. Our findings demonstrate that supplementing bee colonies with sugar syrup and essential oils/probiotic products affects not only the quantity of bee venom collected [32] but also its quality.

#### 3.1.3. Mineral Content

Following the stimulation feeding of the bee colonies, potassium emerged as the macro element with the highest concentration in the analyzed venom samples, ranging from 2.427 mg/g to 10.275 mg/g (Table 4). Calcium followed with values between 0.900 mg/g and 4.650 mg/g, and magnesium ranged from 0.281 mg/g to 0.609 mg/g. Phosphorus had lower concentrations, ranging between 0.235 mg/g and 5.247 mg/g. The determined values for the content of macroelements (Ca, K, Mg, P) indicated the presence of significant differences at α = 0.05, both within the same harvest stage and between different harvest stages. Similar results were found in a study made in three regions in Morocco where potassium was found to be the major macroelement present in our bee venom samples, with a concentration of around 1.62–3.57 mg/g, followed by calcium, sodium, zinc, and magnesium [54].

Potassium remained the predominant macro element in venom samples collected after the rapeseed harvest, with quantities varying from 2.964 mg/g to 8.289 mg/g. Calcium also had a significant presence, ranging from 0.983 mg/g to 2.205 mg/g, followed by magnesium, which ranged from 0.343 mg/g to 0.482 mg/g. Phosphorus values were between 0.104 mg/g and 0.699 mg/g.

After the acacia harvest, potassium continued to be the most abundant macro element, with values ranging from 3.142 mg/g to 8.320 mg/g. Calcium ranged from 0.900 mg/g to 3.747 mg/g, and magnesium from 0.287 mg/g to 0.634 mg/g. Lower amounts of phosphorus were found in all samples, ranging from 0.273 mg/g to 5.799 mg/g.

Thus, the macro element content of the venom harvested after feeding with all three types of food showed that potassium was the most abundant macro element, followed by calcium, magnesium, and phosphorus. Studies conducted by Sabo et al., 2024 [58] on venom samples from various areas in Slovakia reported calcium values of 366.06–1140.06 mg/kg, potassium values of 1221.45–4005.03 mg/kg, and magnesium values of 201.41–948.42 mg/kg. The differences observed in our results may be attributed to the region where the bees are located, as variations in soil and plant composition can affect the mineral makeup of their venom. Additionally, the diet of the bees, including the types of flowers they visit, the nectar they consume, and the types of food supplements provided, along with factors such as temperature, humidity, and overall environmental conditions, can influence the mineral content of their venom [59].

The highest proportion of microelements in the venom analyzed after stimulation feeding ranged between 0.707 mg/g and 1.992 mg/g, representing the zinc content (Table 5). Iron was the second most abundant element, with concentrations between 0.198 mg/g and 1.739 mg/g. Manganese and copper were present in similar quantities, with manganese content ranging from 4.00 µg/g to 36.84 µg/g and copper content ranging from 1.03 µg/g to 18.60 µg/g. The highest values of Fe (V24—1.739 ± 0.119 mg/kg), Mn (V1—36.84 ± 3.28 µg/g), Cu (V22—18.60 ± 0.67 µg/g), and Zn (V7—1.992 ± 0.228 mg/g) contents, determined in venom harvests I, II, and III, differ significantly at a statistical confidence level of α = 0.05. Our findings align with those of El Mehdi et al. [54], who determined the average zinc content in bee venom to be 1.3 mg/g, copper between 3.95 µg/g and 24.72 µg/g, and manganese between 0.78 µg/g and 4.46 µg/g.

After the rapeseed harvest, the venom samples contained zinc in amounts ranging from 0.897 mg/g to 1.586 mg/g, with iron being the second most abundant element (0.361–0.691 mg/g), followed by copper (7.00–11.9 µg/g) and manganese (5.00–9.70 µg/g).

In venom samples collected after acacia harvesting, zinc was the most abundant element, with values ranging from 0.656 mg/g to 1.899 mg/g. Also, high quantities of iron (0.368–1.739 mg/g), copper (5.60–37.4 µg/g), and manganese (4.00–10.20 µg/g) were determined.

In all the samples analyzed, zinc was found in the highest proportion, followed by iron. The soils in the area where the hives were located have an average to good supply of mobile zinc, facilitating the absorption of this element in plants. Consequently, the pollen and nectar of plants in the area have a higher zinc content. For samples collected after stimulation feeding and rapeseed harvesting, copper was the third most abundant element, followed by manganese. The same order of microelements is maintained after acacia harvesting.

These results indicate that feeding bee colonies influences the mineral content of bee venom. Following the evaluation of bee venom collected from different areas of Slovakia, ref. [58] identified the presence of iron (58.02–537.14 mg/kg), manganese (2.27–60.85 mg/kg), copper (1.56–16.86 mg/kg), and zinc (39.68–3305 mg/kg). Choinska et al. [29] determined in bee venom samples from Czech Republic 0.85–1.52 mg Zn/g and 12.2–25.7 µg Cu/g.

In analyzed bee venom samples, Ni and Cd were not detected in any of the three harvests. Pb was identified in samples collected after stimulation feeding, with amounts ranging from 15.34 µg/g to 32.17 µg/g. After the rapeseed harvest, Pb levels ranged from 20.21 µg/g to 44.75 µg/g, and after the acacia harvest, Pb levels were between 23.50 µg/g and 56.05 µg/g (Table 6). These findings exceed those reported by [54], who determined a lead content of 3.24–9.85 µg/g in bee venom samples from Morocco, and by [29], who found 3.04–6.29 mg/kg Pb in bee venom from the Czech Republic. However, they are similar to those of Sabo et al. (2024) [58], who reported Pb values for bee venom from Slovakia ranging from 0.0009 mg/kg to 68.67 mg/kg.

Lead levels in the soils of the studied area ranged from 0.78 ppm to 10.58 ppm, with higher concentrations closer to urban centers and industrial areas. These sources of lead include rock weathering, the use of Pb-containing fertilizers or pesticides, and improper waste disposal. The relatively high lead content is easily absorbed by plants and transferred to bee products. The highest Pb concentrations were found in samples from the third harvest when bees fed on acacia pollen and nectar. Acacia has a root system that exploits a larger amount of soil compared to rapeseed and a longer life cycle, which allows it to absorb higher amounts of Pb. The Pb content in the stimulation feeding variants may come from the pollen and nectar of spontaneous flora, considering that the bees began their flights in mid-March, and the samples were collected at the end of April.

Analyzing the matrix of Pearson correlations (Figure 2), significant correlations are observed at a statistical degree of assurance α = 0.05 in the case of the following determined parameters: Cu-Fe (R^2^ = 0.85), Pb-Fe (R^2^ = 0.72), Pb-Cu (R^2^ = 0.74), Ca-Fe (R^2^ = 0.80), Ca-Cu (R^2^ = 0.86), Ca-Mg (R^2^ = 0.84), and P-Ca (R^2^ = 0.81), with synergisms and antagonisms between elements influencing the mineral composition of bee venom.

### 3.2. Antioxidant Activity by DPPH Method

To evaluate the radical scavenging activity using the DPPH method, we prepared five different concentrations of each extract (2.00 mg/mL, 1.25 mg/mL, 1 mg/mL, 0.625 mg/mL, and 0.313 mg/mL) for the 24 samples tested. Simultaneously, we measured the antioxidant activity of five ascorbic acid solutions at varying concentrations (0.02–0.1 mg/mL) used as a positive control, with the highest concentration (0.1 mg/mL) achieving an inhibition of 91.75%. The IC_50_, representing the concentration required for each extract to achieve 50% inhibition of DPPH, was calculated and expressed in mg/mL.

For the three harvests, the maximum DPPH radical scavenging activity in all analyzed samples was recorded at a concentration of 2.00 mg/mL, with sample V6 having the highest value at 87.05%. The ascorbic acid concentrations used for comparison ranged from 0.02 to 0.1 mg/mL, with the highest inhibition percentage at 0.1 mg/mL being 91.75%, a value higher than that recorded in the venom samples. In every case, the inhibition percentage for ascorbic acid was higher than the maximum value recorded for the venom samples.

At a concentration of 2.00 mg/mL, samples V3, V6, and V15 showed inhibition percentages greater than 80%. Samples V7 and V23 had inhibition percentages below 50%, while the remaining samples ranged between 50% and 80%. The inhibition percentage was directly proportional to the venom concentration, decreasing as the concentration decreased (Table 7).

At a concentration of 1.25 mg/mL, the highest inhibition percentage was 67.90% in sample V15. Six of the 24 samples had inhibition activity greater than 50%, while the other samples were below this value. At a concentration of 1 mg/mL, the highest inhibition activity was 55.36% in sample V15. Samples V7, V8, V11, V16, and V23, tested at this concentration, had inhibition percentages below 30%.

Studies by Gîlcescu Florescu et al. (2024) [60] on *Apis mellifera* bee venom samples collected from the Dolj area of Romania, analyzed at a concentration of 1 mg/mL, showed inhibition percentages of 54.2% and 65.45%. Another study reported inhibition percentages ranging widely from 5.59% to 98% at concentrations of 0.1, 1, 10, and 100 mg/mL [53].

At a concentration of 0.625 mg/mL, sample V15 had the highest inhibition percentage (34.81%), with 10 samples showing percentages below 20%. In a similar study, inhibition percentages at a concentration of 0.6 mg/mL were 35.6% and 45.12% [52]. The lowest concentration analyzed was 0.313 mg/mL, with sample V15 showing maximum activity at 16.85%, a value below the minimum recorded for ascorbic acid. Eight samples had inhibition percentages not exceeding 10% at this concentration.

At a concentration of 2.00 mg/mL, sample V6 exhibited the highest antioxidant activity. However, for the other analyzed concentrations, the maximum inhibition percentage was observed in sample V15. Our research findings align with those of other authors [52,53,57], demonstrating that the antioxidant activity of bee venom is influenced by nutritional factors, specifically the food sources available to the bees. The antioxidant activity of bee venom is influenced by the bees’ diet, which includes nectar and pollen from various plants. Plants rich in antioxidants, such as polyphenols and flavonoids, can enhance the venom’s antioxidant properties. Bees that forage on a diverse range of plants tend to produce venom with higher antioxidant activity due to the variety of bioactive compounds they consume. Seasonal changes in plant availability also affect the nutritional intake of bees and the antioxidant activity of their venom [61].

Table 8 presents the IC_50_, SD, R^2^, and Hill Slope values for the collected venom samples, compared to the values recorded for ascorbic acid used as a standard in the comparison. For ascorbic acid, the IC_50_ value is 2.47 mg/mL, which is lower than the concentrations identified in the venom samples, ranging from 2.87 mg/mL (V15) to 5.76 mg/mL (V23).

The R^2^ value for ascorbic acid is 0.9916, with samples V9 (0.9948) and V15 (0.9956) recording higher values. The Hill Slope value for ascorbic acid is 15.610, whereas samples V3, V5, V6, and V15 have higher values, with sample V6 having the maximum value of 17.609.

Higher Hill Slope values indicate steeper transitions between concentration and effect, suggesting a cooperative mechanism of action where multiple active molecules in the venom work together to produce an amplified effect. Sample V6, having the maximum value, may contain a combination of peptides or other components contributing to more intense antioxidant activity or superior free radical neutralization capacity compared to ascorbic acid. This highlights the potential of bee venom as a source of potent bioactive compounds.

Even though sample V15 did not have the highest DPPH value at the highest extract concentration, its low IC_50_ value suggests that it is more effective at scavenging free radicals at lower concentrations. Therefore, sample V15 can be considered the sample with the best antioxidant capacity among the ones we analyzed.

Figure 3 illustrates that Principal Component 1 (PC1) captures 29.25% of the total variance in the dataset, indicating that nearly one-third of the data’s variability is attributed to this component. Principal Component 2 (PC2) accounts for 21.12% of the total variance, representing roughly one-fifth of the data’s variability. Together, PC1 and PC2 explain approximately 50.37% of the total variance, providing a substantial overview of the dataset in a two-dimensional space.

In the PCA plot, the vectors for IC_50_ and DPPH2 are oriented in the same direction, suggesting a positive correlation. This implies that samples with higher IC 50 values also exhibit higher DPPH2 values. Given that both variables are associated with antioxidant capacity, their similar positioning on the plot indicates that samples with higher concentrations of these antioxidants show better scavenging activity. Essentially, higher IC_50_ values correlate with higher DPPH2 values, reflecting a strong positive relationship.

The vectors for Fe and Impurities point in opposite directions, indicating a negative correlation. This means that as the iron content in a sample increase, the impurity levels tend to decrease, and vice versa. This negative correlation helps us understand how iron and impurity levels vary inversely across different samples, which means that samples rich in iron are likely to have lower impurities.

The antioxidant capacity of bee venom is primarily attributed to several bioactive substances, including melittin, the main component of bee venom with strong antioxidant properties, and phospholipase A2 (PLA2). Various other peptides and enzymes present in bee venom also contribute to its overall antioxidant capacity. Bobis et al., 2017 [62] demonstrated that minerals such as zinc (Zn), iron (Fe), copper (Cu), and manganese (Mn) possess antioxidant properties and can enhance the overall antioxidant capacity of bee venom. However, our study, as depicted in Figure 3, indicates that mineral elements do not influence the antioxidant capacity of bee venom.

## 4. Conclusions

Since the composition of bee venom is influenced by their diet, as well as other factors such as temperature, humidity, and general environmental conditions, it can vary from one region to another. Therefore, the values obtained in our study slightly differ from those presented in another available research. Samples collected in April (stimulation feeding) showed the lowest moisture values in the case of using basil essential oil and probiotics, and the highest values were obtained in the case of control groups. The pH values were higher compared to those reported in other studies, indicating possible differences related to venom composition influenced by food type and harvesting period. Impurities showed high variability between samples, and statistically significant differences between variants of a harvesting session and between harvests indicate that they are influenced by the harvesting type. The absence of a clear standard regarding the permissible level of impurities shows the need for studies to establish quality limits.

The analysis of the venom harvested after feeding with all three types of food revealed that potassium was the most abundant macro element, followed by calcium, magnesium and phosphorus. Zinc was the predominant microelement in all samples, with iron, copper, and manganese following in varying orders depending on the season. The observed differences in our results may be attributed to the region where the bees are located, as variations in soil and plant composition can influence the mineral content of their venom. Among the heavy metals analyzed, Pb was determined in high quantities in bee venom, which may be an indicator of Pb pollution in the studied area.

The DPPH radical scavenging activity of the analyzed bee venom was significantly influenced by the concentration used, with the inhibition percentage increasing proportionally to the sample concentration. The highest antioxidant activity was recorded at a concentration of 2.00 mg/mL. Although this value is remarkable, it is lower than the percentage inhibition of ascorbic acid used for comparison. Only three samples showed inhibition percentages higher than 80% at this concentration, while other samples had values ranging between 50% and 80%, and two samples recorded inhibitions below 50%.

Sample feed with rapeseed nectar and pollen (V15) can be considered the sample with the best antioxidant capacity among the ones we analyzed.

Investigating how stimulatory feeding impacts the mineral content and antioxidant properties of bee venom holds importance for both human and bee welfare. For humans, understanding the venom’s composition and how it changes is crucial in improving diagnostic methods and treatments for allergies caused by bee stings. Additionally, bee venom plays a therapeutic role in addressing conditions such as arthritis, multiple sclerosis, and chronic pain, making it vital to analyze its mineral makeup for safer and more effective applications. The unique mineral properties of the venom may even contribute to the development of innovative pharmaceuticals.

For bees, venom serves as an essential defense mechanism, and its composition ensures its potency against predators. Furthermore, the process of venom production reflects the overall health of bees, providing valuable insights into environmental influences affecting their populations. By studying these factors, we can gain a deeper understanding of their survival and well-being.

## Figures and Tables

**Figure 1 insects-16-00423-f001:**
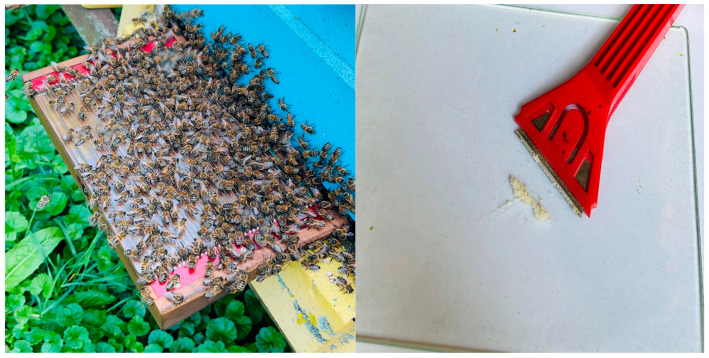
BeeWhisper v.5.1 collector model 2016.

**Figure 2 insects-16-00423-f002:**
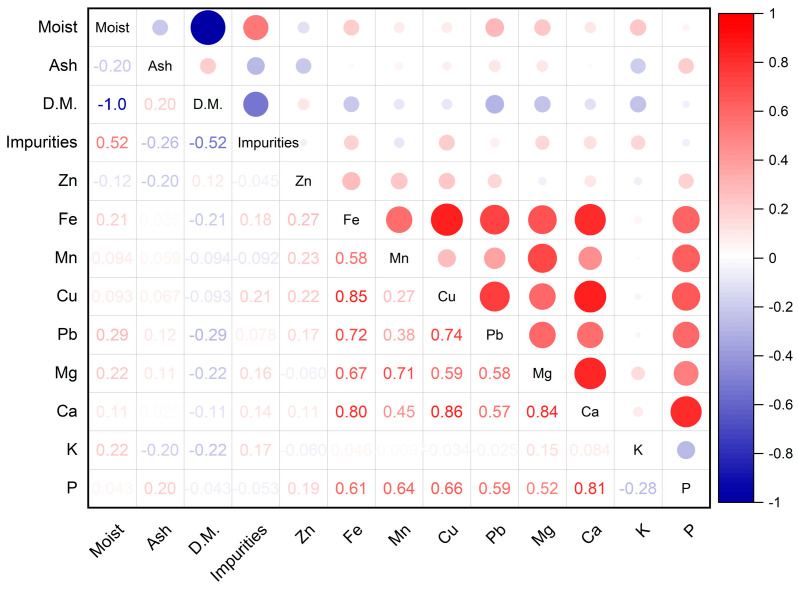
Pearson correlation matrix.

**Figure 3 insects-16-00423-f003:**
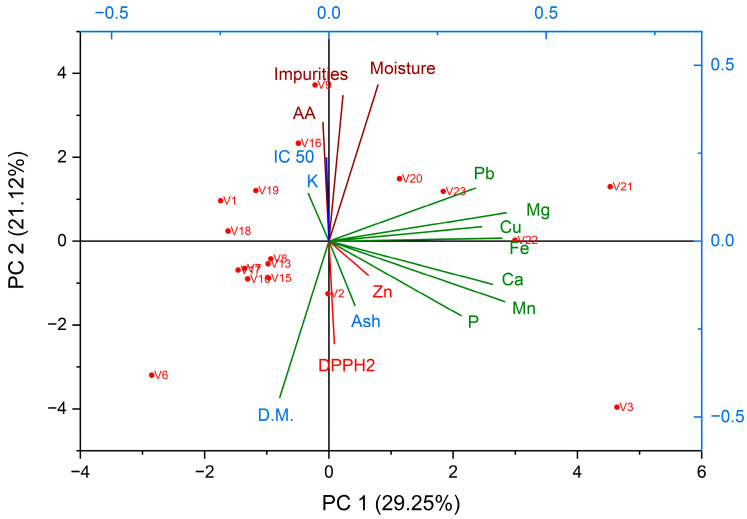
Principal component analysis boxplot.

**Table 1 insects-16-00423-t001:** Experiment organization scheme.

Venom Sample	Harvest Period	Food Source	Abbreviation
1	April	Sugar syrup 1 L	V1
2	April	Sugar syrup 1 L + basil essential oil 30 µL	V2
3	April	Sugar syrup 1 L + thyme essential oil 30 µL	V3
4	April	Sugar syrup 1 L + oregano essential oil 30 µL	V4
5	April	Sugar syrup 1 L + Lacium 1 capsule	V5
6	April	Sugar syrup 1 L + Colobiotic 1 capsule	V6
7	April	Sugar syrup 1 L + Enterolactis Plus 1 capsule	V7
8	April	Sugar syrup 1 L + Lacium 1 capsule + oregano essential oil 30 µL	V8
9	May	Rapeseed nectar and pollen	V9
10	May	Rapeseed nectar and pollen	V10
11	May	Rapeseed nectar and pollen	V11
12	May	Rapeseed nectar and pollen	V12
13	May	Rapeseed nectar and pollen	V13
14	May	Rapeseed nectar and pollen	V14
15	May	Rapeseed nectar and pollen	V15
16	May	Rapeseed nectar and pollen	V16
17	June	Acacia nectar and pollen	V17
18	June	Acacia nectar and pollen	V18
19	June	Acacia nectar and pollen	V19
20	June	Acacia nectar and pollen	V20
21	June	Acacia nectar and pollen	V21
22	June	Acacia nectar and pollen	V22
23	June	Acacia nectar and pollen	V23
24	June	Acacia nectar and pollen	V24

**Table 2 insects-16-00423-t002:** Working conditions for Varian Spectra 240 FS spectrophotometer.

Metal	λ (nm)	Lamp Current (mA)	Slit Width (mm)
Ni	232	4	0.2
Cr	357.9	7	0.2
Cu	324.8	4	0.5
Cd	228.8	4	0.5
Mn	279.5	5	0.2
Zn	213	5	1
Fe	248.3	5	0.2
Ca	422.7	10	0.5
Mg	285.2	4	0.5
Pb	283.3	10	1.2
K	414 nm	4	0.5

**Table 3 insects-16-00423-t003:** Moisture content, dry matter, ash, impurities, pH, and total amino acids of bee venom.

		**Venom Sample Harvest I**								
		**V1**	**V2**	**V3**	**V4**	**V5**	**V6**	**V7**	**V8**	** *p* **
Humidity (%)		18.00 ± 2.00 ^a,A,B^	13.00 ± 2.65 ^a,A^	15.00 ± 1.73 ^a,A^	17.00 ± 1.73 ^a,A^	14.00 ± 2.00 ^a,A^	13.00 ± 2.65 ^a,A,B^	14.00 ± 2.00 ^a,A,B^	15.00 ± 0.00 ^a,A^	0.065
DM (%)		82.00 ± 2.00 ^a,A,B^	87.00 ± 2.65 ^a,A^	85.00 ± 1.73 ^a,A^	83.00 ± 1.73 ^a,A,B^	86.00 ± 2.00 ^a,A^	87.00 ± 2.65 ^a,A,B^	86.00 ± 2.00 ^a,A^	85.00 ± 0.00 ^a,A^	0.065
pH		6.41 ± 0.02 ^a,A^	5.95 ± 0.04 ^b,l,A^	6.10 ± 0.05 ^c,f,A^	6.02 ± 0.05 ^b,c,f,g,h,A^	5.96 ± 0.04 ^b,g,h,A^	5.84 ± 0.05 ^g,i,k,l^	5.81 ± 0.04 ^d,i,j,A,B^	5.96 ± 0.06 ^e,h,k,l,A,B^	0.000
Impurities (%)		13.33 ± 0.15 ^a,A^	10.00 ± 0.20 ^b,I,A^	6.67 ± 0.15 ^c,A^	10.00 ± 0.10 ^d,i,A^	10.00 ± 0.15 ^e,i,A^	6.67 ± 0.08 ^c,f,A^	9.96 ± 0.14 ^b,g,i,A^	10.00 ± 0.05 ^h,I,A^	0.000
Total amino acids (mg/g)		398.20 ± 1.37 ^a,A^	368.17 ± 1.67 ^b,A^	335.95 ± 1.49 ^c,A^	346.33 ± 1.94 ^d,A^	351.24 ± 1.25 ^e,A^	362.16 ± 1.86 ^f,A^	359.98 ± 1.65 ^g,A^	350.15 ± 1.75 ^h,A^	0.000
Ash (%)		5.70 ± 0.07 ^a,A^	3.00 ± 0.11 ^b,A^	4.50 ± 0.03 ^c,A^	4.00 ± 0.04 ^d,A^	4.80 ± 0.08 ^e,A^	6.70 ± 0.02 ^f,A^	5.80 ± 0.05 ^a,A^	4.10 ± 0.05 ^g,A^	0.000
		**Venom Sample Harvest II**								
		**V9**	**V10**	**V11**	**V12**	**V13**	**V14**	**V15**	**V16**	
Humidity (%)		22.00 ± 2.00 ^a,A^	16.00 ± 3.00 ^a,c,A^	14.00 ± 0.00 ^b,c,A^	16.00 ± 4.00 ^a,c,A^	14.00 ± 1.73 ^b,c,A^	10.00 ± 3.61 ^b,c,A^	17.00 ± 1.00 ^a,c,A^	18.00 ± 1.73 ^a,c,B^	0.002
DM (%)		78.00 ± 2.00 ^a,A^	84.00 ± 3.00 ^a,c,A^	86.00 ± 0.00 ^b,c,A^	84.00 ± 4.00 ^a,c,A^	86.00 ± 1.73 ^b,c,A^	90.00 ± 3.61 ^b,c,A^	83.00 ± 1.00 ^a,c,A,B^	82.00 ± 1.73 ^a,c,B^	0.002
pH		6.11 ± 0.04 ^a,B^	5.94 ± 0.04 ^b,A^	6.11 ± 0.04 ^a,A^	5.90 ± 0.05 ^b,c,A^	5.71 ± 0.04 ^f,B^	5.54 ± 0.04 ^f,B^	5.85 ± 0.04 ^b,c,A^	5.93 ± 0.06 ^b,c,A^	0.000
Impurities (%)		20.00 ± 0.15 ^A,B^	10.00 ± 0.08 ^b,i,A^	13.33 ± 0.08 ^c,B^	6.67 ± 0.10 ^d,i,B^	6.67 ± 0.09 ^e,i,B^	10.00 ± 0.13 ^c,f,B^	6.67 ± 0.08 ^b,g,i,B^	13.33 ± 0.11 ^h,i,B^	0.000
Total amino acids (mg/g)		373.63 ± 1.91 _a,B_	354.52 ± 1.86 _b,i,B_	354.52 ± 1.86 _c,i,B_	362.16 ± 2.10 _d,B_	384.00 ± 2.13 _e,B_	363.25 ± 1.75 _f,B_	381.27 ± 1.75 _g,B_	403.11 ± 1.25 ^h,B^	0.000
Ash (%)		3.31 ± 0.04 ^a,B^	7.40 ± 0.05 ^b,B^	3.31 ± 0.08 ^a,B^	7.90 ± 0.05 ^c,B^	3.50 ± 0.04 ^d,B^	6.60 ± 0.04 ^e,B^	3.60 ± 0.04 ^d,f,B^	1.50 ± 0.05 ^g,B^	0.000
		**Venom Sample Harvest III**								
		**V17**	**V18**	**V19**	**V20**	**V21**	**V22**	**V23**	**V24**	
Humidity (%)		15.00 ± 1.73 ^a,B^	16.00 ± 3.00 ^a,A^	17.00 ± 2.00 ^a,A^	17.00 ± 0.00 ^a,A^	18.00 ± 0.00 ^a,B^	17.00 ± 1.00 ^a,B^	18.00 ± 1.00 ^a,B^	17.00 ± 0.00 ^a,A,B^	0.308
DM (%)		85.00 ± 1.73 ^a,B^	84.00 ± 3.00 ^a,A^	83.00 ± 2.00 ^a,A^	83.00 ± 0.00 ^a,B^	82.00 ± 0.00 ^a,B^	83.00 ± 1.00 ^a,B^	82.00 ± 1.00 ^a,B^	83.00 ± 0.000 ^a,A,B^	0.308
pH		5.98 ± 0.06 ^a,C^	6.04 ± 0.04 ^a,B^	5.87 ± 0.05 ^b,c,B^	5.94 ± 0.06 ^a,c,A^	5.95 ± 0.05 ^a,c,A^	6.03 ± 0.04 ^a,C^	5.93 ± 0.05 ^a,c,B^	6.06 ± 0.04 ^a,B^	0.001
Impurities (%)		7.00 ± 0.10 ^a,C^	6.67 ± 0.12 ^a,B^	13.33 ± 0.18 ^b,c,e,B^	13.33 ± 0.13 ^b,c,e,C^	13.33 ± 0.21 ^b,c,e,C^	10.00 ± 0.09 ^d,B^	6.67 ± 0.10 ^a,B^	13.33 ± 0.18 ^e,B^	0.000
Total amino acids (mg/g)		407.48 ± 1.750 ^a,C^	396.56 ± 1.752 ^b,C^	397.65 ± 1.617 ^c,C^	394.92 ± 1.834 ^d,C^	396.02 ± 0,436 ^e,i,C^	396.02 ± 1.601 ^f,i,C^	400.93 ± 2.265 ^g,C^	403.66 ± 1.931 ^h,C^	0.000
Ash (%)		2.00 ± 0.07 ^a,e,C^	2.30 ± 0.04 ^a,e,C^	2.40 ± 0.03 ^a,C^	1.91 ± 0.05 ^e,C^	5.30 ± 0.05 ^b,C^	6.20 ± 0.04 ^c,C^	6.04 ± 0.47 ^c,A^	4.60 ± 0.04 ^d,C^	0.000
*p*	Humidity	0.012	0.398	0.125	0.857	0.028	0.047	0.031	0.027	-
	DM	0.012	0.398	0.125	0.857	0.028	0.047	0.031	0.027	-
	pH	0.000	0.024	0.001	0.076	0.001	0.000	0.027	0.048	-
	Impurities	0.000	0.000	0.000	0.000	0.000	0.000	0.000	0.000	-
	Total Aa	0.000	0.000	0.000	0.000	0.000	0.000	0.000	0.000	-
	Ash	0.000	0.000	0.000	0.000	0.000	0.000	0.000	0.000	-

Means marked with the same letters show no statistically significant differences (*p* > 0.05), while means associated with different letters demonstrate statistically significant differences (*p* < 0.05). Lowercase letters are used to denote statistically significant differences between groups within the same harvest, whereas uppercase letters indicate statistically significant differences between harvests. Harvest I after stimulation feeding (V1—Sugar syrup; V2—Sugar syrup and basil essential oil; V3—Sugar syrup and thyme essential oil; V4—Sugar syrup and oregano essential oil; V5—Sugar syrup and Lacium; V6—Sugar syrup and Colobiotic; V7—Sugar syrup and Enterolactis Plus; V8—Sugar syrup, oregano essential oil and Lacium); Harvest II (V9–V16 after rapeseed harvest); Harvest III (V17–V24 after acacia harvest).

**Table 4 insects-16-00423-t004:** Macroelement content of bee venom.

		**Venom Sample Harvest I**								
		**V1**	**V2**	**V3**	**V4**	**V5**	**V6**	**V7**	**V8**	** *p* **
Ca (mg/g)		1.010 ± 0.090 ^a,A^	3.588 ± 0.190 ^b,A^	4.650 ± 0.150 ^c,A^	1.463 ± 0.120 ^d,A^	1.506 ± 0.110 ^e,A^	1.076 ± 0.100 ^f,A^	1.270 ± 0.270 ^g,A^	1.432 ± 0.230 ^h,A^	0.000
K (mg/g)		8.362 ± 0.360 ^a,A^	10.275 ± 0.300 ^b,A^	3.659 ± 0.460 ^c,A^	2.427 ± 0.170 ^d,A^	3.588 ± 0.210 ^e,A^	3.576 ± 0.130 ^f,A^	5.17 ± 0.310 ^g,A^	4.828 ± 0.190 ^h,A^	0.000
Mg (mg/g)		0.365 ± 0.023 ^a,A^	0.523 ± 0.016 ^b,A^	0.609 ± 0.022 ^c,A^	0.386 ± 0.014 ^d,A^	0.427 ± 0.017 ^e,A^	0.281 ± 0.019 ^f,A^	0.367 ± 0.026 ^g,A^	0.412 ± 0.012 ^h,A^	0.000
P (mg/g)		0.994 ± 0.027 ^a,A^	0.235 ± 0.015 ^b,A^	0.524 ± 0.024 ^c,A^	-	-	1.105 ± 0.010 ^d,A^	0.524 ± 0.026 ^e,A^	0.354 ± 0.016 ^f,A^	0.000
		**Venom Sample Harvest II**								
		**V9**	**V10**	**V11**	**V12**	**V13**	**V14**	**V15**	**V16**	
Ca (mg/g)		1.684 ± 0.037 ^a,B^	0.983 ± 0.058 ^b,B^	2.205 ± 0.095 ^c,B^	1.410 ± 0.075 ^d,B^	1.162 ± 0.102 ^e,B^	1.353 ± 0.026 ^f,B^	1.217 ± 0.097 ^g,B^	1.3817 ± 0.078 ^h,B^	0.000
K (mg/g)		8.289 ± 0.239 ^a,B^	6.25 ± 0.162 ^b,B^	3.426 ± 0.116 ^c,B^	4.536 ± 0.176 ^d,B^	7.929 ± 0.171 ^e,B^	3.143 ± 0.143 ^f,B^	5.032 ± 0.278 ^g,B^	2.964 ± 0.146 ^h,B^	0.000
Mg (mg/g)		0.477 ± 0.020 ^a,B^	0.346 ± 0.024 ^b,B^	0.482 ± 0.018 ^c,B^	0.477 ± 0.027 ^d,B^	0.431 ± 0.025 ^e,B^	0.402 ± 0.045 ^f,B^	0.343 ± 0.030 ^g,B^	0.416 ± 0.027 ^h,B^	0.000
P (mg/g)		0.428 ± 0.028 ^a,B^	0.104 ± 0.015 ^b,B^	-	-	0.530 ± 0.065 ^c,B^	-	0.335 ± 0.039 ^d,B^	0.699 ± 0.022 ^e,B^	0.000
		**Venom Sample Harvest III**								
		**V17**	**V18**	**V19**	**V20**	**V21**	**V22**	**V23**	**V24**	
Ca (mg/g)		0.900 ± 0.064 ^a,C^	1.166 ± 0.150 ^b,C^	1.087 ± 0.123 ^c,C^	2.676 ± 0.218 ^d,C^	2.185 ± 0.550 ^e,C^	3.747 ± 0.297 ^f,C^	3.231 ± 0.131 ^g,C^	0.992 ± 0.092 ^h,C^	0.000
K (mg/g)		5.423 ± 0.295 ^a,C^	7.746 ± 0.314 ^b,C^	3.142 ± 0.142 ^c,C^	8.320 ± 0.270 ^d,C^	6.248 ± 0.248 ^e,C^	3.931 ± 0.431 ^f,C^	5.574 ± 0.224 ^g,C^	-	0.000
Mg (mg/g)		0.287 ± 0.036 ^a,C^	0.292 ± 0.018 ^b,C^	0.316 ± 0.018 ^c,C^	0.593 ± 0.027 ^d,C^	0.604 ± 0.016 ^e,C^	0.574 ± 0.023 ^f,C^	0.634 ± 0.034 ^g,C^	-	0.000
P (mg/g)		0.556 ± 0.046 ^a,C^	0.313 ± 0.023 ^b,C^	0.289 ± 0.033 ^c,C^	0.383 ± 0.024 ^d,C^	1.980 ± 0.280 ^e,C^	0.273 ± 0.033 ^f,C^	2.611 ± 0.111 ^g,C^	5.799 ± 0.239 ^h,C^	0.000
*p*	Ca	0.000	0.000	0.000	0.000	0.000	0.000	0.000	0.000	-
	K	0.000	0.000	0.000	0.000	0.000	0.000	0.000	0.000	-
	Mg	0.000	0.000	0.000	0.000	0.000	0.000	0.000	0.000	-
	P	0.000	0.000	0.000	0.000	0.000	0.000	0.000	0.000	-

Means marked with the same letters show no statistically significant differences (*p* > 0.05), while means associated with different letters demonstrate statistically significant differences (*p* < 0.05). Lowercase letters are used to denote statistically significant differences between groups within the same harvest, whereas uppercase letters indicate statistically significant differences between harvests. Harvest I after stimulation feeding (V1—Sugar syrup; V2—Sugar syrup and basil essential oil; V3—Sugar syrup and thyme essential oil; V4—Sugar syrup and oregano essential oil; V5—Sugar syrup and Lacium; V6—Sugar syrup and Colobiotic; V7—Sugar syrup and Enterolactis Plus; V8—Sugar syrup, oregano essential oil and Lacium); Harvest II (V9–V16 after rapeseed harvest); Harvest III (V17–V24 after acacia harvest).

**Table 5 insects-16-00423-t005:** Microelement content of bee venom.

		**Venom Sample Harvest I**								
		**V1**	**V2**	**V3**	**V4**	**V5**	**V6**	**V7**	**V8**	** *p* **
Fe (mg/g)		0.308 ± 0.016 ^a,A^	0.694 ± 0.038 ^b,h,A^	0.988 ± 0.078 ^c,A^	0.458 ± 0.038 ^d,A^	0.470 ± 0.038 ^e,i,A^	0.198 ± 0.006 ^a,A^	0.479 ± 0.048 ^f,i,A^	0.585 ± 0.046 ^g,h,i,A^	0.000
Mn (µg/g)		36.84 ± 3.28 ^a,A^	10.14 ± 0.37 ^b,g,A^	29.651 ± 1.47 ^c,A^	6.00 ± 0.11 ^d,h,A^	5.50 ± 0.32 ^d,e,i,A^	4.40 ± 0.21 ^a,A^	6.08 ± 0.63 ^f,h,i,A^	9.72 ± 0.36 ^g,A^	0.000
Cu (µg/g)		4.40 ± 0.18 ^a,A^	1.10 ± 0.10 ^b,c,A^	1.11 ± 0.22 ^c,e,A^	4.90 ± 0.22 ^d,A^	4.90 ± 0.34 ^b,e,A^	4.30 ± 0.51 ^a,A^	9.30 ± 0.53 ^f,A^	9.00 ± 0.48 ^f,g,A^	0.000
Zn (mg/g)		1.561 ± 0.082 ^a,A^	0.751 ± 0.056 ^b,d,g,A^	1.958 ± 0.143 ^c,A^	0.894 ± 0.065 ^d,g,A^	1.613 ± 0.108 ^a,A^	0.707 ± 0.056 ^b,d,e,A^	1.992 ± 0.228 ^c,f,A^	1.041 ± 0.030 ^g,A^	0.000
		**Venom Sample Harvest II**								
		**V9**	**V10**	**V11**	**V12**	**V13**	**V14**	**V15**	**V16**	
Fe (mg/g)		0.508 ± 0.041 ^a,f,B^	0.467 ± 0.050 ^a,f,B^	0.691 ± 0.036 ^b,d,B^	0.627 ± 0.036 ^c,d,e,B^	0.361 ± 0.028 ^f,B^	0.450 ± 0.033 ^a,f,B^	0.521 ± 0.071 ^a,e,A^	0.495 ± 0.026 ^a,A^	0.000
Mn (µg/g)		9.20 ± 0.31 ^a,B^	6.90 ± 0.31 ^b,f,g,B^	9.70 ± 0.42 ^c,B^	8.60 ± 0.28 ^d,B^	5.00 ± 0.56 ^e,h,B^	6.60 ± 0.46 ^f,B^	7.10 ± 0.54 ^g,B^	5.10 ± 0.22 ^h,B^	0.000
Cu (µg/g)		7.00 ± 0.47 ^a,B^	9.20 ± 0.84 ^b,f,B^	1.03 ± 0.14 ^c,e,B^	1.19 ± 0.32 ^d,B^	1.05 ± 0.16 ^e,B^	9.00 ± 0.83 ^f,g,B^	8.70 ± 0.40 ^g,B^	9.21 ± 0.29 ^b,f,h,A^	0.000
Zn (mg/g)		0.897 ± 0.051 ^a,B^	1.114 ± 0.073 ^a,f,B^	1.356 ± 0.093 ^b,e,f,B^	1.010 ± 0.080 ^a,A^	1.371 ± 0.103 ^c,f,g,A^	1.365 ± 0.090 ^d,f,B^	1.586 ± 0.129 ^e,g,B^	1.064 ± 0.106 ^a,A^	0.000
		**Venom Sample Harvest III**								
		**V17**	**V18**	**V19**	**V20**	**V21**	**V22**	**V23**	**V24**	
Fe (mg/g)		0.479 ± 0.026 ^a,B^	0.718 ± 0.062 ^b,e,A^	0.368 ± 0.033 ^a,C^	0.737 ± 0.057 ^b,B^	1.405 ± 0.044 ^c,C^	1.290 ± 0.046 ^c,C^	0.513 ± 0.033 ^a,e,A^	1.739 ± 0.119 ^d,B^	0.000
Mn (µg/g)		7.70 ± 0.32 ^a,C^	7.50 ± 0.38 ^a,C^	4.00 ± 0.20 ^b,C^	10.50 ± 0.48 ^c,C^	18.30 ± 0.32 ^d,C^	11.70 ± 0.64 ^e,C^	12.70 ± 0.47 ^f,C^	10.20 ± 0.47 ^g,C^	0.000
Cu (µg/g)		9.10 ± 0.22 ^a,C^	5.60 ± 0.40 ^b,C^	10.01 ± 0.32 ^c,B^	9.40 ± 0.61 ^d,C^	18.60 ± 0.67 ^e,C^	14.90 ± 0.94 ^f,C^	8.40 ± 0.41 ^g,C^	3.74 ± 0.27 ^h,B^	0.000
Zn (mg/g)		1.899 ± 0.078 ^a,C^	1.480 ± 0.041 ^b,f,C^	1.483 ± 0.160 ^c,f,B^	1.701 ± 0.067 ^a,f,B^	1.469 ± 0.204 ^d,f,A^	1.540 ± 0.215 ^a,f,B^	0.656 ± 0.075 ^e,C^	1.549 ± 0.126 ^a,f,B^	0.000
*p*	Fe	0.000	0.000	0.000	0.001	0.000	0.000	0.618	0.000	-
	Mn	0.000	0.000	0.000	0.000	0.000	0.000	0.000	0.000	-
	Cu	0.000	0.000	0.000	0.000	0.000	0.000	0.000	0.000	-
	Zn	0.000	0.000	0.000	0.000	0.206	0.001	0.000	0.001	-

Means marked with the same letters show no statistically significant differences (*p* > 0.05), while means associated with different letters demonstrate statistically significant differences (*p* < 0.05). Lowercase letters are used to denote statistically significant differences between groups within the same harvest, whereas uppercase letters indicate statistically significant differences between harvests. Harvest I after stimulation feeding (V1—Sugar syrup; V2—Sugar syrup and basil essential oil; V3—Sugar syrup and thyme essential oil; V4—Sugar syrup and oregano essential oil; V5—Sugar syrup and Lacium; V6—Sugar syrup and Colobiotic; V7—Sugar syrup and Enterolactis Plus; V8—Sugar syrup, oregano essential oil and Lacium); Harvest II (V9–V16 after rapeseed harvest); Harvest III (V17–V24 after acacia harvest).

**Table 6 insects-16-00423-t006:** Toxic metal content of bee venom.

	**Venom Sample Harvest I**								
	**V1**	**V2**	**V3**	**V4**	**V5**	**V6**	**V7**	**V8**	** *p* **
Ni (µg/g)	Undetectable	Undetectable	Undetectable	Undetectable	Undetectable	Undetectable	Undetectable	Undetectable	-
Cd (µg/g)	Undetectable	Undetectable	Undetectable	Undetectable	Undetectable	Undetectable	Undetectable	Undetectable	-
Pb (µg/g)	23.90 ± 1.10 ^a,A^	18.19 ± 0.91 ^b,f,A^	32.17 ± 0.92 ^c,A^	15.34 ± 0.43 ^d,A^	21.09 ± 0.64 ^e,A^	18.27 ± 0.72 ^f,A^	18.77 ± 0.77 ^g,A^	20.48 ± 0.51 ^h,A^	0.000
	**Venom Sample Harvest II**								
	**V9**	**V10**	**V11**	**V12**	**V13**	**V14**	**V15**	**V16**	
Ni (µg/g)	Undetectable	Undetectable	Undetectable	Undetectable	Undetectable	Undetectable	Undetectable	Undetectable	-
Cd (µg/g)	Undetectable	Undetectable	Undetectable	Undetectable	Undetectable	Undetectable	Undetectable	Undetectable	-
Pb (µg/g)	20.21 ± 1.26 ^a,B^	24.36 ± 0.51 ^b,f,B^	28.72 ± 1.77 ^c,B^	44.75 ± 1.25 ^d,B^	35.12 ± 1.27 ^e,B^	24.45 ± 1.20 ^f,B^	27.16 ± 0.99 ^g,B^	33.94 ± 1.94 ^h,B^	0.000
	**Venom Sample Harvest III**								
	**V17**	**V18**	**V19**	**V20**	**V21**	**V22**	**V23**	**V24**	
Ni (µg/g)	Undetectable	Undetectable	Undetectable	Undetectable	Undetectable	Undetectable	Undetectable	Undetectable	-
Cd (µg/g)	Undetectable	Undetectable	Undetectable	Undetectable	Undetectable	Undetectable	Undetectable	Undetectable	-
Pb (µg/g)	25.80 ± 1.20 ^a,B^	23.50 ± 1.11 ^b,B^	32.48 ± 1.63 ^c,B^	36.24 ± 1.04 ^d,B^	53.22 ± 1.11 ^e,B^	38.43 ± 1.23 ^f,B^	38.83 ± 1.73 ^g,B^	56.05 ± 0.75 ^h,B^	0.000
*p*	0.000	0.000	0.000	0.000	0.000	0.000	0.000	0.000	-

Means marked with the same letters show no statistically significant differences (*p* > 0.05), while means associated with different letters demonstrate statistically significant differences (*p* < 0.05). Lowercase letters are used to denote statistically significant differences between groups within the same harvest, whereas uppercase letters indicate statistically significant differences between harvests. Harvest I after stimulation feeding (V1—Sugar syrup; V2—Sugar syrup and basil essential oil; V3—Sugar syrup and thyme essential oil; V4—Sugar syrup and oregano essential oil; V5—Sugar syrup and Lacium; V6—Sugar syrup and Colobiotic; V7—Sugar syrup and Enterolactis Plus; V8—Sugar syrup, oregano essential oil and Lacium); Harvest II (V9–V16 after rapeseed harvest); Harvest III (V17–V24 after acacia harvest).

**Table 7 insects-16-00423-t007:** DPPH radical scavenging activity (% inhibition) of aqueous extracts compared to ascorbic acid.

**Concentration (mg/mL)**	**Venom Sample Harvest I**								**Ascorbic Acid**	
	**V1**	**V2**	**V3**	**V4** **(% Inhibition)**	**V5**	**V6**	**V7**	**V8**	**Concentration (mg/mL)**	**(% Inhibition)**
2.00	73.91 ± 2.11 ^b^	67.14 ± 2.06 ^c,d^	82.62 ± 1.57 ^a^	69.24 ± 4.45 ^b,c^	79.40 ± 6.23 ^a^	87.05 ± 1.07 ^a^	43.31 ± 1.31 ^i^	50.07 ± 1.82 ^g,h^	0.10	91.75
1.25	49.89 ± 0.9 ^e–g^	45.84 ± 1.23 ^h,i^	56.39 ± 1.39 ^b,c^	43.52 ± 1.65 ^i,j^	54.58 ± 1.11 ^c,d^	59.36 ± 1.48 ^b,c^	36.59 ± 1.34 ^m,n^	40.08 ± 1.67 ^k,l^	0.08	71.59
1.00	41.47 ± 0.88 ^a–f^	31.75 ± 1.07 ^d–g^	42.75 ± 1.59 ^a–e^	34.13 ± 1.78 ^c–g^	41.95 ± 0.71 ^a–f^	46.65 ± 1.37 ^a–c^	24.14 ± 1.69 ^g^	28.05 ± 1.36 ^f,g^	0.06	56.02
0.625	25.78 ± 15.57 ^a–c^	23.55 ± 0.68 ^bc^	19.09 ± 1.08	23.94 ± 1.62 ^bc^	28.24 ± 1.11 ^bc^	28.51 ± 1.27 ^bc^	18.56 ± 0.76 ^bc^	18.75 ± 1.10 ^c^	0.04	44.94
0.313	13.97 ± 1.13 ^b–e^	11.61 ± 0.73 ^f–i^	14.62 ± 0.93 ^a–c^	11.94 ± 1.07 ^e–h^	13.89 ± 1.14 ^b–e^	14.43 ± 0.69 ^b–d^	6.74 ± 0.62 ^m^	8.93 ± 0.72 ^k–m^	0.02	27.02
	**Venom Sample Harvest II**									
	**V9**	**V10**	**V11**	**V12**	**V13**	**V14**	**V15**	**V16**		
2.00	60.15 ± 1.04 ^e^	71.46 ± 1.21 ^b,c^	59.89 ± 0.65 ^e,f^	70.39 ± 0.64 ^b,c^	62.04 ± 0.72 ^d,e^	66.98 ± 1.87 ^c,d^	86.53 ± 1.06 ^a^	53.88 ± 0.93 ^f,g^	0.10	91.75
1.25	47.67 ± 0.78 ^g,h^	48.18 ± 0.82 ^f–h^	38.42 ± 0.96 ^l,m^	45.93 ± 0.95 ^hi^	39.03 ± 0.80 ^l,m^	43.34 ± 0.78 ^i,j^	67.90 ± 0.94 ^a^	34.15 ± 0.91 ^n^	0.08	71.59
1.00	37.08 ± 0.95 ^b–g^	44.14 ± 0.95 ^a–d^	28.77 ± 0.84 ^e–g^	36.74 ± 0.87 ^b–g^	31.30 ± 0.51 ^d–g^	34.32 ± 0.65 ^c–g^	55.36 ± 1.13 ^a^	27.91 ± 0.67 ^a–c^	0.06	56.02
0.625	20.57 ± 10.12 ^c^	26.11 ± 9.89 ^a–c^	17.84 ± 5.84 ^a,b^	22.58 ± 7.45 ^b,c^	19.91 ± 6.89 ^a–c^	18.83 ± 9.32 ^b,c^	34.81 ± 11.25 ^a–c^	17.57 ± 5.63 ^a^	0.04	44.94
0.313	10.75 ± 0.43 ^g–k^	14.28 ± 0.32 ^b–d^	8.39 ± 0.61 ^l–m^	9.39 ± 0.44 ^i–l^	9.19 ± 0.70 ^j–l^	9.20 ± 0.59 ^i–l^	16.85 ± 0.54 ^a–c^	9.58 ± 0.58 ^i–l^	0.02	27.02
	**Venom Sample Harvest III**									
	**V17**	**V18**	**V19**	**V20**	**V21**	**V22**	**V23**	**V24**		
2.00	74.47 ± 1.51 ^b^	59.78 ± 0.54 ^e,f^	66.34 ± 0.94 ^c,d^	61.52 ± 0.58 ^d,e^	66.84 ± 1.16 ^c,d^	71.90 ± 0.55 ^d,e^	47.52 ± 0.67 ^g–i^	74.97 ± 0.39 ^b^	0.10	91.75
1.25	49.61 ± 0.39 ^e–g^	41.51 ± 0.48 ^j–l^	39.82 ± 0.30 ^k,l^	42.63 ± 0.72 ^j–l^	42.42 ± 0.56 ^j–l^	50.91 ± 0.41 ^e–g^	30.15 ± 0.85 ^o^	52.66 ± 0.43 ^d,e^	0.08	71.59
1.00	40.03 ± 0.73 ^b–f^	32.43 ± 0.58 ^d–g^	31.21 ± 0.77 ^d–g^	36.71 ± 20.89 ^d–g^	34.17 ± 5.26 ^b–f^	41.65 ± 0.43 ^a–f^	23.63 ± 0.37 ^g^	41.09 ± 0.48 ^b–f^	0.06	56.02
0.625	26.16 ± 1.29 ^b,c^	19.50 ± 1.50 ^b,c^	20.66 ± 0.34 ^b,c^	19.72 ± 0.82 ^b,c^	24.00 ± 0.86 ^b,c^	25.86 ± 0.44 ^b,c^	16.37 ± 0.63 ^c^	28.68 ± 0.68 ^b,c^	0.04	44.94
0.313	12.84 ± 0.72 ^c–g^	12.46 ± 0.53 c–g	8.93 ± 0.28 ^k–m^	12.24 ± 0.74 ^d–h^	11.21 ± 0.79 ^g–j^	13.59 ± 0.81 ^b–f^	10.13 ± 0.24 ^h–l^	15.29 ± 0.97 ^a,b^	0.02	27.02

Results are expressed as the three-determination mean ± standard deviation (SD). Means marked with the same letters show no statistically significant differences (*p* > 0.05), while means associated with different letters demonstrate statistically significant differences (*p* < 0.05). Lowercase letters are used to denote statistically significant differences between groups within the same harvest. Harvest I after stimulation feeding (V1—Sugar syrup; V2—Sugar syrup and basil essential oil; V3—Sugar syrup and thyme essential oil; V4—Sugar syrup and oregano essential oil; V5—Sugar syrup and Lacium; V6—Sugar syrup and Colobiotic; V7—Sugar syrup and Enterolactis Plus; V8—Sugar syrup, oregano essential oil and Lacium); Harvest II (V9–V16 after rapeseed harvest); Harvest III (V17–V24 after acacia harvest).

**Table 8 insects-16-00423-t008:** IC_50_ values of samples compared to ascorbic acid.

	**Venom Sample Harvest I**								
	**V1**	**V2**	**V3**	**V4**	**V5**	**V6**	**V7**	**V8**	**Ascorbic Acid**
IC_50_ SD (mg/mL)	3.62 ± 0.0755	4.05 ± 0.1323	3.30 ± 0.0300	4.00 ± 0.1100	3.41 ± 0.1015	3.16 ± 0.0819	5.65 ± 0.1323	5.01 ± 0.0854	2.47 ± 0.0458
R^2^	0.9759	0.9693	0.9782	0.9537	0.9812	0.9814	0.9878	0.9983	0.9916
Hill Slope	14.399	13.335	16.330	13.418	15.736	17.609	9.117	10.361	15.610
	**Venom Sample Harvest II**								
	**V9**	**V10**	**V11**	**V12**	**V13**	**V14**	**V15**	**V16**	
IC_50_ SD (mg/mL)	4.17 ± 0.0656	3.68 ± 0.0557	4.56 ± 0.0656	3.89 ± 0.0600	4.42 ± 0.0721	4.10 ± 0.0917	2.87 ± 0.0721	5.03 ± 0.1572	2.47 ± 0.0458
R^2^	0.9948	0.9675	0.9675	0.9758	0.9627	0.9719	0.9956	0.9588	0.9916
Hill Slope	12.590	13.605	12.358	14.535	12.482	14.007	17.245	10.518	15.610
	**Venom Sample Harvest III**								
	**V17**	**V18**	**V19**	**V20**	**V21**	**V22**	**V23**	**V24**	
IC_50_ SD (mg/mL)	3.64 ± 0.0656	4.45 ± 0.1323	4.24 ± 0.0656	4.27 ± 0.0656	4.10 ± 0.1000	3.65 ± 0.1000	5.76 ± 0.1015	3.52 ± 0.0819	2.47 ± 0.0458
R^2^	0.9749	0.9762	0.9487	0.9739	0.9589	0.9869	0.9454	0.9827	0.9916
Hill Slope	14.671	11.665	13.398	12.147	12.968	14.167	8.856	14.334	15.610

Results are expressed as the three-determination mean ± standard deviation (SD). Harvest I after stimulation feeding (V1—Sugar syrup; V2—Sugar syrup and basil essential oil; V3—Sugar syrup and thyme essential oil; V4—Sugar syrup and oregano essential oil; V5—Sugar syrup and Lacium; V6—Sugar syrup and Colobiotic; V7—Sugar syrup and Enterolactis Plus; V8—Sugar syrup, oregano essential oil and Lacium); Harvest II (V9–V16 after rapeseed harvest); Harvest III (V17–V24 after acacia harvest).

## Data Availability

The original contributions presented in the study are included in the article. Further inquires can be directed to the corresponding authors.

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
