# Peer review of "Assessing the Influence of Stimulatory Feeding of Bee Colonies on Mineral Composition and Antioxidant Activity of Bee Venom"

_insects, 2025, doi:10.3390/insects16040423_

Round 1

Reviewer 1 Report

Comments and Suggestions for Authors

The authors have presented a unique study examining the impacts of diets on bee venom composition. I have minor suggestions. There are numerous typos throughout the manuscript. Please check as edit as needed. Adding recent important publications (such as Toxicological study of bee venom (Apis mellifera mellifera) from different regions of the province of Buenos Aires, Argentina) and cross references will improve the introduction and discussion. A section is needed to explain thoroughly why understanding this change is important and why these results are relevant from both honey bee and human perspectives. 

Author Response

University: University of Life Sciences “King Mihai I” from Timișoara, Calea Aradului no.119

Address: Calea Aradului no.119, 300641, Timișoara, România

Journal: Insects (ISSN 2075-4450)

Manuscript ID: insects-3537492

Type: Article

Title: Assessing the influence of stimulatory feeding of bee colonies on mineral composition and antioxidant activity of bee venom  

Authors: Adrian Dan Rășinar, Isidora Radulov*, Adina Berbecea*, Doris Floares (Oarga), Nicoleta Vicar, Eliza Simiz, Monica Dragomirescu, Silvia Pătruică*

Section: Social Insects and Apiculture

Reviewer

The authors have presented a unique study examining the impacts of diets on bee venom composition. I have minor suggestions. There are numerous typos throughout the manuscript. Please check as edit as needed. Adding recent important publications (such as Toxicological study of bee venom (Apis mellifera mellifera) from different regions of the province of Buenos Aires, Argentina) and cross references will improve the introduction and discussion. A section is needed to explain thoroughly why understanding this change is important and why these results are relevant from both honey bee and human perspectives.

Dear Reviewer,

Thank you for your thorough review of our paper and for your appreciation of the data presented on bee venom from Timiș County, Romania. We highly value your suggestions and have implemented the minor corrections you indicated to ensure the manuscript is as clear and precise as possible.

We have revised the manuscript according to your recommendations and believe that the changes made enhance the clarity and accuracy of the presented information. We have added the requested section and marked it in green. We hope that the updated version meets your expectations and can be accepted for publication.

Once again, we sincerely appreciate your valuable contribution!

Best regards,
The Authors

Reviewer 2 Report

Comments and Suggestions for Authors

The authors present a study on mineral composition of bee venom and how different feeds influence this composition as well as bee venoms antioxidant activity. This is a very interesting study that should be published. Especially considering that there is not a lot of data on bee venom element composition. Some changes should be made before accepting the manuscript for publication.

Some specific comments:

Abstract:

L29: I would use the term macro and micro-elements and be consistent throughout the manuscript.

L34: Please rephrase, the sentence seems incomplete.  

Introduction:

L60-67: I would suggest that this is moved after paragraph ending with L45.

Materials and Methods:

L710-114: Please rephrase.

L177: Please provide more details. You have 48 containers with 24 samples. You tested three (L178) of each sample?

Results:

3.1.3. Mineral content

The main comment for this section is the lack of statistical analyses between different stimulation feeding recipes and then between different times of sampling/harvest. Giving only a range of concentrations measured and concluding that there are differences should be backed by statistical analyses (ANOVA with post hoc).  Although later I saw the statistical results are present in the table, it would be better to have them in the text and mention which samples are statistically different between themselves. Also, mention which had the highest/lowest concentrations of which micro and macro elements.

L248-249: In Table 1. Colobiotic is designated V6, while here in the text you say V5.

L250-251: Sample V14 is mentioned as “Colobiotic”. Why? Did bees keep receiving probiotics during the May and June harvests? If not, why do you mention Colobiotic? Is this the same colonies that received Colobiotic during the May harvest?

L330: Should it be macro-element?

L355-363: When comparing to other studies it would be useful to use the same units. Either by concerting your values to mg/kg or the other studies to mg/g. As it is now someone could misinterpret it that the differences are different hundreds of orders of magnitude. The same applies throughout the manuscript.

L369: Use proper in text citation style.

3.2. Antioxidant Activity by DPPH method

L478-480: It would be interesting to elaborate on this some more.

 L489-493: I must admit I am not an expert on antioxidant activity. However, here it is stated that LC50 is 2.47 mg/mL for ascorbic acid, while at the beginning of this section you state that at 0.1mg/mL 91.75% inhibition was observed? This is a huge discrepancy. The same applies to all other samples tested.

L489: Please define SEM and R2.

Conclusion

L544-546: influenced by harvest type

L564: I would suggest saying which sample it is (which harvest/supplement feed), just a number is not very informative to readers.

Author Response

University: University of Life Sciences “King Mihai I” from Timișoara, Calea Aradului no.119

Address: Calea Aradului no.119, 300641, Timișoara, România

Journal: Insects (ISSN 2075-4450)

Manuscript ID: insects-3537492

Type: Article

Title: Assessing the influence of stimulatory feeding of bee colonies on mineral composition and antioxidant activity of bee venom  

Authors: Adrian Dan Rășinar, Isidora Radulov*, Adina Berbecea*, Doris Floares (Oarga), Nicoleta Vicar, Eliza Simiz, Monica Dragomirescu, Silvia Pătruică*

Section: Social Insects and Apiculture

Reviewer

The authors present a study on mineral composition of bee venom and how different feeds influence this composition as well as bee venoms antioxidant activity. This is a very interesting study that should be published. Especially considering that there is not a lot of data on bee venom element composition. Some changes should be made before accepting the manuscript for publication.

Some specific comments:

Abstract:

L29: I would use the term macro and micro-elements and be consistent throughout the manuscript.

A: Thank you for the observation. We have made the change.

L34: Please rephrase, the sentence seems incomplete. 

A: Thank you for the observation. We have rephrased and included it: “In all samples, the highest DPPH radical scavenging activity was observed at a con-centration of 2.00 mg/mL, with sample V6 showing the maximum value of 87.05%.“

Introduction:

L60-67: I would suggest that this is moved after paragraph ending with L45.

A: Thank you for the suggestion, we have made the change and we’ve marked it in red.

Materials and Methods

L110-114: Please rephrase.

A: Thank you for the observation. We have rephrased: “The essential oils used in the study were purchased from Bioinovativ (https://life-bio.ro/produse/uleiuri-esentiale/page/6 / ). They were obtained through steam distillation and are certified by the Romanian National Service for Medicinal, Aromatic Plants and Beehive Products. The oils included thyme whole essential oil, wild oregano/sorrel whole essential oil, and basil whole essential oil. “

L177: Please provide more details. You have 48 containers with 24 samples. You tested three (L178) of each sample?

A: Thank you for the observation. We have included it to make things clearer: “ To determine the amino acids, 72 lidded containers were prepared corresponding to 3 determinations for each of the 24 experimental variants.“

Results:

3.1.3. Mineral content

The main comment for this section is the lack of statistical analyses between different stimulation feeding recipes and then between different times of sampling/harvest. Giving only a range of concentrations measured and concluding that there are differences should be backed by statistical analyses (ANOVA with post hoc).  Although later I saw the statistical results are present in the table, it would be better to have them in the text and mention which samples are statistically different between themselves. Also, mention which had the highest/lowest concentrations of which micro and macro elements.

A: You're right, we included it in the text.

L248-249: In Table 1. Colobiotic is designated V6, while here in the text you say V5.

A: You are right, we have corrected it.

L250-251: Sample V14 is mentioned as “Colobiotic”. Why? Did bees keep receiving probiotics during the May and June harvests? If not, why do you mention Colobiotic? Is this the same colonies that received Colobiotic during the May harvest?

A: Thank you for the observation. The bees did not continue to receive probiotics in May. We mentions Colobiotics because these are the same colonies that received Colobiotic in April.

L330: Should it be macro-element?

A: We have made the change.

L355-363: When comparing to other studies it would be useful to use the same units. Either by concerting your values to mg/kg or the other studies to mg/g. As it is now someone could misinterpret it that the differences are different hundreds of orders of magnitude. The same applies throughout the manuscript.

A: Throughout the paper, we used mg/kg as the unit of measurement for macro elements and µg/g to correlate with the quantity of the element found in venom. In the scientific literature, data regarding the amount of macro and micro elements can be found expressed in mg/kg (Sabo, R.; Staroň, M.; Sabová, L.; Jančo, I.; Tomka, M.; Árvay, J. Toxic and essential elements in honeybee venom from Slovakia: Potential health risk to humans. Heliyon 2024, 10, e39282, doi:https://doi.org/10.1016/j.heliyon.2024.e39282), mg/g (El Mehdi, I.; Falcão, S.I.; Harandou, M.; Boujraf, S.; Calhelha, R.C.; Ferreira, I.C.F.R.; Anjos, O.; Campos, M.G.; Vilas-Boas, M. Chemical, Cytotoxic, and Anti-Inflammatory Assessment of Honey Bee Venom from Apis mellifera intermissa. Antibiotics 2021, 10, 1514.), or µg/g (El Mehdi, I.; Falcão, S.I.; Harandou, M.; Boujraf, S.; Calhelha, R.C.; Ferreira, I.C.F.R.; Anjos, O.; Campos, M.G.; Vilas-Boas, M. Chemical, Cytotoxic, and Anti-Inflammatory Assessment of Honey Bee Venom from Apis mellifera intermissa. Antibiotics 2021, 10, 1514.).

L369: Use proper in text citation style.

A: We have corrected it and used the proper format.

3.2. Antioxidant Activity by DPPH method

L478-480: It would be interesting to elaborate on this some more.

A: We have elaborated and added to the text the influence of the food source on the antioxidant activity of the venom.

 L489-493: I must admit I am not an expert on antioxidant activity. However, here it is stated that LC50 is 2.47 mg/mL for ascorbic acid, while at the beginning of this section you state that at 0.1mg/mL 91.75% inhibition was observed? This is a huge discrepancy. The same applies to all other samples tested.

A: Indeed, it may seem like a discrepancy, but this can be explained by the fact that the IC50 value is derived from a mathematical regression based on the entire data set, and the dose-response curve is not necessarily linear. In the case of compounds with high antioxidant activity, maximum inhibition may be achieved at very low concentrations, resulting in IC50 values that can appear counterintuitive. Some antioxidant molecules may form aggregates or interact with free radicals in a manner that diminishes efficiency once the concentration surpasses a certain threshold.

L489: Please define SEM and R2.

A: Indeed, it was a typing error. We have corrected it, thank you.

 SEM = SD standard deviation

R2 (R-squared) = is a statistical measure that indicates how well the data fits a regression model.

Conclusion

L544-546: influenced by harvest type

A: We have made the correction.

L564: I would suggest saying which sample it is (which harvest/supplement feed), just a number is not very informative to readers.

A: : We have made the correction.

We have revised the manuscript according to your recommendations and believe that the changes made enhance the clarity and accuracy of the presented information.

We are grateful for your time and effort to review our paper and we hope we have successfully addressed all your queries!

Sincerely,

The authors.